# RETROSPECTION: LEVERAGING THE PAST FOR EFFICIENT TRAINING OF DEEP NEURAL NETWORKS

## ABSTRACT

Deep neural networks are powerful learning machines that have enabled breakthroughs in several domains. In this work, we introduce retrospection loss to improve performance of neural networks by utilizing prior experiences during training. Minimizing the retrospection loss pushes the parameter state at the current training step towards the optimal parameter state while pulling it away from the parameter state at a previous training step. We conduct extensive experiments to show that the proposed retrospection loss results in improved performance across multiple tasks, input types and network architectures.

## 1 INTRODUCTION

Large deep neural networks have enabled breakthroughs in fields such as computer vision (Krizhevsky et al., 2012), speech recognition (Hinton et al., 2012), natural language understanding (Mikolov et al., 2013) and reinforcement learning (Mnih et al., 2015). Hence, in recent times, significant effort has been directed towards enhancing network efficiency through data augmentation, regularization methods and novel training strategies (Zhong et al., 2017) (Zhang et al., 2017), (Huang et al., 2017) (Noh et al., 2017), (Wang et al., 2018) (Han et al., 2016). In this work, we introduce a technique to improve performance by utilizing prior experiences of the network during training.

Humans are efficient learners with the ability to quickly understand and process diverse ideas. A hallmark of human intelligence is the capability to internalize these complex ideas by actively referencing past interpretations to continually adapt understanding. Our artificial agents should be able to do the same, learning and adapting quickly. This kind of fast and flexible learning is challenging, since the agent must effectively integrate its prior experience with a small amount of new information, while avoiding overfitting to the new data.

In this work, we introduce a new retrospection loss that utilizes prior training experiences of a deep neural network to guide parameter updates and improve performance. The idea for the retrospection loss is simple - to ensure that the predictions at a training step are more similar to the ground truth than to the predictions from a previous training step. As training proceeds, minimizing the loss constrains the network parameters to continually evolve towards the optimal state by successive constriction of the predictions into tighter spaces around the goal. The proposed retrospection loss is simple, easy to implement and we empirically show that it works well across multiple tasks, input types and network architectures.

The key contributions of our work can be summarized as:

- We propose a new simple, easy to implement retrospective loss that is based on looking back at the trajectory of gradient descent and providing an earlier parameter state as guidance for further learning.

- We exhaustively experiment on a wide range of tasks including image classification (+ few-shot learning), GANs, speech recognition, text classification and consistently beat state-of-the-art methods on benchmark datasets with the addition of this loss term.

- To the best of our knowledge, this is the first such effort; our empirical studies showed a consistent improvement in performance across the tasks in our multiple trials, demonstrating the potential of this method to have a strong impact on practical use in real-world applications across domains.

## 2 RELATED WORK

The retrospection loss leverages the parameter state from a previous training step as guidance to compute the current gradient update. Correspondingly, one could find similarities with efforts in optimization, that utilize information from past training steps for future weight updates as well as methods that leverage guidance from other parameter states during training.

Techniques such as SVRG (Johnson & Zhang, 2013), SARAH(Nguyen et al., 2017), ProxSARAH (Pham et al., 2019) use gradients from earlier training steps to predict better weight updates. Other optimization methods like Momentum (Sutskever et al., 2013), Adam (Kingma & Ba, 2014) Nesterov Momentum (Jin et al., 2018) accumulate past gradients to accelerate weight updates in the right direction in order to achieve faster convergence. In contrast, our work introduces an additional training objective to guide convergence, and can be used to improve performance when used with different optimizer configurations, as shown in our results.

In reinforcement learning (RL), where techniques involve optimizing using moving (evolving) targets, methods for Q-learning and policy gradients benefit from using a guidance network during training. The DQN algorithm proposed by (Mnih et al., 2015) uses an additional target network (same as online network) for Q-value updates, where parameters are updated by copying from the online network at discrete steps. Double Q-learning (Hasselt, 2010) learns two Q functions, where each Q-function is updated with a value for the next state from the other Q-function. Policy gradient methods such as TRPO (Schulman et al., 2015), PPO (Schulman et al., 2017) use a KL-divergence objective during training that constrains the loss to ensure deviation from a previously learned policy is small. In these techniques, leveraging a guidance during training results in improved convergence and sample efficiency. Note that all these efforts are constrained to the RL setting. Further, the objective in the RL setting is to control divergence from the guidance step to better handle moving targets. On the other hand, the proposed retrospection loss is formalized differently to address the supervised learning setting. To the best of our knowledge, this is the first such effort that uses an idea such as retrospection in supervised learning.

## 3 METHODOLOGY

We now present the formulation of our retrospective loss. Consider a neural network, $g(\cdot)$, parameterized by its weights $\theta$. Let the optimal parameters of the neural networks at the end of training be given by $\theta^*$. The current parameters of the network at time step $T$ during training are given by $\theta^T$. The objective of the retrospective loss is to leverage the past states during training, and cue the network to be closer to the ground truth than a past state at time step $T_p$. Given an input data-label pair $(\mathbf{x}_i, y_i)$, the retrospective loss is given by:

$$\mathcal{L}_{retrospective}^T = \kappa * ||g_{\theta^T}(\mathbf{x}_i) - y_i|| - ||g_{\theta^T}(\mathbf{x}_i) - g_{\theta^{T_p}}(\mathbf{x}_i)|| \tag{1}$$

The retrospective loss is designed such that minimizing it with respect to $\theta$ over the training steps would constrain the parameter state at each reference step $\theta^T$ to be more similar to $\theta^*$ than the parameter state from the delayed time step $\theta^{T_p}$. The $\kappa$ scaling term is required to obtain sufficient gradient signal in later stages of training when $g_{\theta^T}(\mathbf{x}_i)$ is close to $y_i$, and the first term becomes small.

Adding this loss term to an existing supervised learning task loss provides for efficient training, as shown in our experiments later. The retrospective loss is introduced to the training objective following a warm-up period wherein the neural network function can be considered stable for use of retrospective updates. The training objective at any training step $T$ with the retrospective loss is hence defined as:

$$\mathcal{L} = \begin{cases} \mathcal{L}_{task} & T < \mathcal{I}_W \\ \mathcal{L}_{task} + \mathcal{L}_{retrospective}^T & T \geq \mathcal{I}_W \end{cases} \tag{2}$$

where $\mathcal{L}_{task}$ is the task-specific training objective and $\mathcal{I}_W$ is the number of warm-up iterations. We simply use $T_p = F * \lfloor T/F \rfloor$ as the time step for retrospection in this work, and show gains in efficiency of training. One could however mine for $T_p$ intelligently to further improve the performance. Here, F is the retrospective update frequency, which gives an upper bound difference of the previous training step (Tp) from the current training step T at which we compute the retrospective loss.

**Geometric Intuition.** Figure 1 illustrates the geometric intuition of the working of the retrospective loss. By design (Eqn 1), $\mathcal{L}_{retrospective}$ is negative when the current parameter state is farther away from the retrospective step, $T_p$, than the optimal solution (which is the desirable objective).

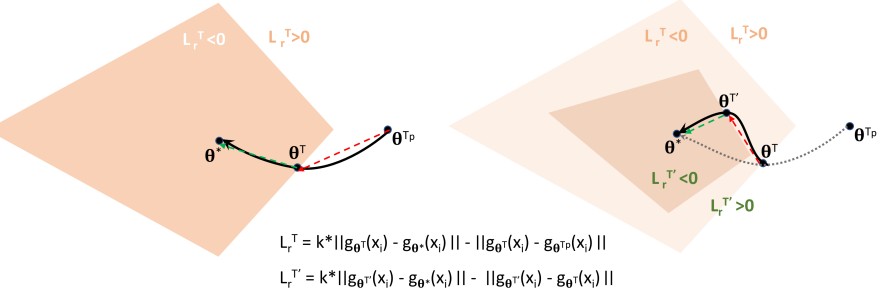

$$L_r^T = k^* ||g_{\theta^T}(x_i) - g_{\theta^*}(x_i)|| - ||g_{\theta^T}(x_i) - g_{\theta^{T_p}}(x_i)||$$

$$L_r^{T'} = k^* ||g_{\theta^{T'}}(x_i) - g_{\theta^*}(x_i)|| - ||g_{\theta^{T'}}(x_i) - g_{\theta^T}(x_i)||$$

Figure 1: Geometric intuition of the working of the proposed retrospection loss. The figures show polytopes in the weight parameter space. *(Left)* For all $\theta^i$ inside the shown colored polytope, the retrospective loss is negative and is positive outside. Our objective is to push parameters of the current $\theta^T$ further inside this polygon close to $\theta^*$; *(Right)* In a future time step $T' > T$, by design of the retrospective loss, the polytope region shrinks and our objective at this time step is to push parameters to a near-optimal region around $\theta^*$.

One could view the loss term as dividing the parameter space into two regions: a polytope around the optimal $\theta^*$ where $\mathcal{L}_{retrospective} < 0$, and the region outside the polytope where $\mathcal{L}_{retrospective} > 0$. Minimizing retrospective loss pushes the network towards parameters further inside the polytope, thus helping speed up the training process. As shown on the right subfigure in Figure 1, the polytope shrinks over time, since the retrospective

---

**Algorithm 1** Retrospective Training

1: **Input**: Training Set V, Current Model Parameters $\theta^T$, Previous State Model Parameters $\theta^{T_p}$, Update Frequency F, # of Warm-Up Iterations $\mathcal{I}_W$,
2: **for** Step 1 to n **do**
3:    $grad_{task} \leftarrow 0$ (Initialising the gradients w.r.t task-specific loss)
4:    $grad_{retrospective} \leftarrow 0$ (Initialising the gradients w.r.t retrospective loss)
5:    Training Data of minibatch size B pairs of $(X(i), Y(i))$.
6:    $L(\theta^T, X(i), Y(i)) = L_{task}(\theta^T(X(i)), Y(i))$
7:    $grad_{task} \leftarrow \nabla(L(\theta^T, X(i), Y(i)))$
8:    **if** $Step > \mathcal{I}_W$ **then**
9:      $L(\theta^T, \theta^{T_p}, X(i), Y(i)) = L_{retrospective}(\theta^T(X(i)), \theta^{T_p}(X(i)), Y(i))$
10:      $grad_{retrospective} \leftarrow \nabla(L(\theta^T, \theta^{T_p}, X(i), Y(i)))$
11:    **end if**
12:    **if** Step % F == 0 **then**
13:      $\theta^{T_p} \leftarrow \theta^T$
14:    **end if**
15:    $\theta^T \leftarrow \theta^T - \eta * (grad_{task} + grad_{retrospective})$
16: **end for**
17:

---

support, $T_p$, is also updated to more recent parameter states. This helps further push the parameters into a near-optimal region around $\theta^*$. The loss term helps in improved solution in most cases, and faster training in certain cases, as shown in our extensive empirical studies in Section 4. Algorithm 1 summarizes the methodology.

**Connection with Triplet Loss.** The triplet loss ((Chechik et al., 2010; Schroff et al., 2015; Hoffer & Ailon, 2015)) has been proposed and used extensively over the last few years to learn high-quality data embeddings, by considering a triplet of data points, $\mathbf{x}_a$ (anchor point), $\mathbf{x}_p$ (point from the positive/same class as the sample under consideration), and $\mathbf{x}_n$ (point from the negative class/class different from the sample under consideration). The loss is then defined as:

$$\max\left(\|g_a - g_p\|^2 - \|g_a - g_n\|^2 + m, 0\right) \tag{3}$$

where $g$ is the neural network model, and $m$ is a minimum desired margin of separation. The triplet loss, inspired by contrastive loss (Hadsell et al., 2006), attempts to learn parameters $\theta$ of a neural network in such a way that data points belonging to the same class are pulled together closer than a data point from another class. One could view the proposed retrospection loss as a triplet loss in the parameter space. While the traditional triplet loss consider a triplet of data samples, we consider a triplet of parameters, $\theta^T$, $\theta^*$, and $\theta^{T_p}$. We however believe that retrospection captures the proposed loss better, since we consider previous parameter states in time.

**Connection with Momentum.** Viewing retrospection from the perspective of previous gradients in the training trajectory, one can connect it to the use of momentum, although more in a contrasting sense. The use of momentum and variants such as Nesterov momentum (Jin et al., 2018) in training neural networks use the past gradient, say at $\theta^{T-1}$ or the gradient over the previous few steps, at $\{\theta^{T-q}, \cdots, \theta^{T-1}\}, q > 0$), while updating the parameters in the current step. This assumes local

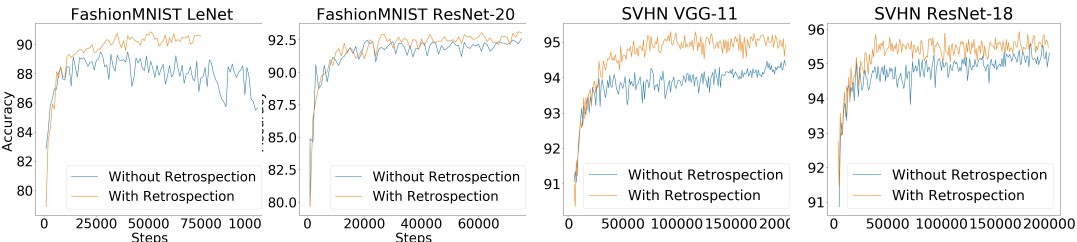

Figure 2: Classification performance using retrospection on F-MNIST and SVHN datasets

consistency of the direction of the gradient update in the training trajectory, and that one can use these previous directions to get a more robust estimate of the gradient step to be taken currently. In contrast, retrospection leverages the same idea from the opposite perspective, viz., consistency of the direction of the gradient update is *only* local, and hence the parameter state, $\theta^{T_p}$ farther away from the current state $\theta^T$, provides a cue of what the next parameter must be far from. This raises interesting discussions, and the possibility of analyzing retrospection as a thrust obtained from an undesirable parameter state, as opposed to momentum. We leave these as interesting directions of future work, and focus this work on proposing the method, and showing its effectiveness in training neural networks.

## 4 EXPERIMENTS AND RESULTS

We conduct experiments using retrospection on the following tasks: image classification (Sec 4.1), image generation (Sec 4.2), speech recognition (Sec 4.3), text classification (Sec 4.4) and few-shot image classification (Sec 4.5). During experimentation, the original (without retrospection) and retrospective (with retrospection) configurations are trained using same weight initialization, to ensure consistency of comparison. For all experiments, we use the $L_1$-norm as the choice of norm in our implementation for the retrospective loss (Eqn 1). When retrospection is used without warm-up, the guidance parameters, $\theta^{T_p}$, are initialized at random.

### 4.1 IMAGE CLASSIFICATION

We perform image classification experiments using Fashion-MNIST (Xiao et al., 2017), SVHN (Netzer et al., 2011) and CIFAR-10 (Krizhevsky, 2009) datasets. The retrospection loss, for classification, uses activations of the softmax layer. The default hyperparameter configurations for retrospection include a warm-up period of zero epochs and a retrospective update frequency of fifty steps. The parameter, $\kappa$, is initialized at 4 and increased by 2% at each retrospective update. Quantitative results for image classification are compiled in Table 1.

**Fashion-MNIST.** For experiments on Fashion MNIST, we use LeNet (Lecun et al., 2001) and ResNet-20 (He et al., 2016) architectures. Models in each experiment are trained to convergence using the SGD optimizer (lr=0.1, momentum=0.5, mini-batch=32) running over 70,000 steps. Results in Figure 2 (a)-(b) show that using the retrospective loss results in improved training.

**SVHN.** For experiments on SVHN, we use VGG-11 (Simonyan & Zisserman, 2014) and ResNet-18 (He et al., 2016) architectures. Models in each experiment are trained to convergence using the SGD optimizer (lr=0.001, momentum=0.9, mini-batch=100) running over 200,000 steps. Results in Figure 2 (c)-(d) show that using the retrospective loss results in more efficient training.

| Dataset | Model | Original | Retrospective |
|---------|-------|----------|---------------|
| F-MNIST | LeNet | 10.8 | **9.4** |
|         | ResNet-20 | 7.6 | **6.8** |
| SVHN | VGG-11 | 5.54 | **4.70** |
|      | ResNet-18 | 4.42 | **4.06** |
| CIFAR-10 | ResNet-44 | 6.98 (7.17) | **6.55** |
|          | ResNet-56 | 6.86 (6.97) | **6.52** |
|          | ResNet-110 | 6.55 (6.61) | **6.29** |

Table 1: Classification error using retrospection on F-MNIST, SVHN and CIFAR-10 dataset

**CIFAR-10.** For experiments on CIFAR-10 (Krizhevsky, 2009), we use larger variants of ResNet including ResNet - 44, 56, 110 (He et al., 2016). Models in each experiment are trained for 200 epochs, using the training configuration (mini-batch, lr policy) detailed in (He et al., 2016). Here, we observe that

using the retrospection loss in later stages of training results in best improvement in performance. Correspondingly, the retrospective loss is introduced after a warm-up of 150 epochs and the retrospective update frequency is one epoch. The parameter, $\kappa$, is initialized at 4 and updated by 2% once every ten retrospective updates. Quantitative performance is reported in Table 1. For sake of completion, we also mention (in brackets) the error rates for the corresponding experiments as reported by authors in the original work (He et al., 2016).

## 4.2 IMAGE GENERATION

Next, we perform experiment with Generative Adversarial Networks (GAN) using Fashion-MNIST (F-MNIST) (Xiao et al., 2017) and CIFAR-10 (Krizhevsky, 2009) datasets. Our study considers both unconditional (DCGAN, LSGAN) and conditional (ACGAN) variants of GANs. We adapt implementations from (dcg) for LSGAN (Mao et al., 2016), DCGAN (Radford et al., 2015) and from (acg) for ACGAN (Odena et al., 2016). For our experiments, we train the generator and discriminator for 100 epochs, with initial learning rate of 0.0002 on minibatches of size 64 using Adam optimizer. We report performance using Inception Score (Salimans et al., 2016), a standard metric for evaluating GANs. The inception score is calculated using implementation in (inc, 2018) with predictions for CIFAR-10 generated using network in (Szegedy et al., 2015) and features for F-MNIST using network in (Krizhevsky et al., 2012).

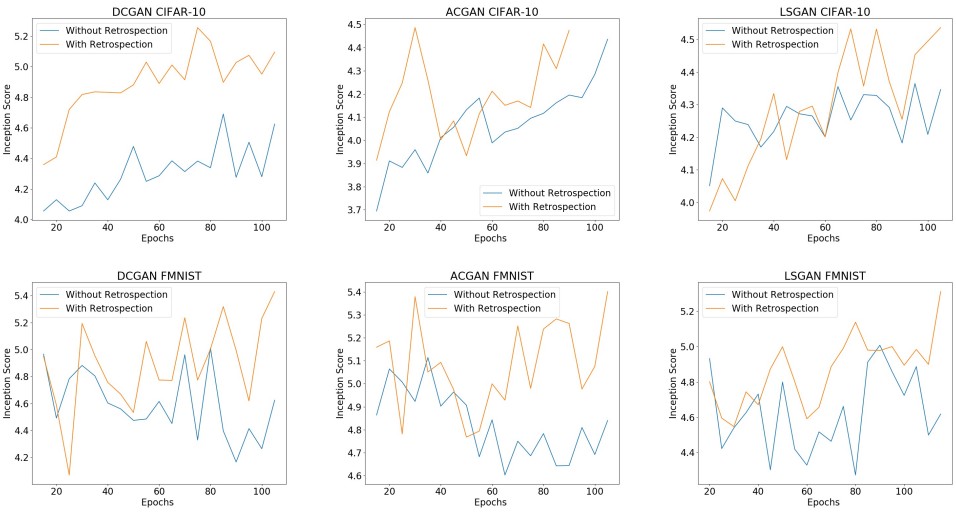

Figure 3: Inception Scores using retrospection on CIFAR-10 (Krizhevsky, 2009)(row 1) and F-MNIST (Xiao et al., 2017) (row 2) datasets using DCGAN (Radford et al., 2015) (col 1), ACGAN (Odena et al., 2016)(col 2), LSGAN (Mao et al., 2016)(col 3).

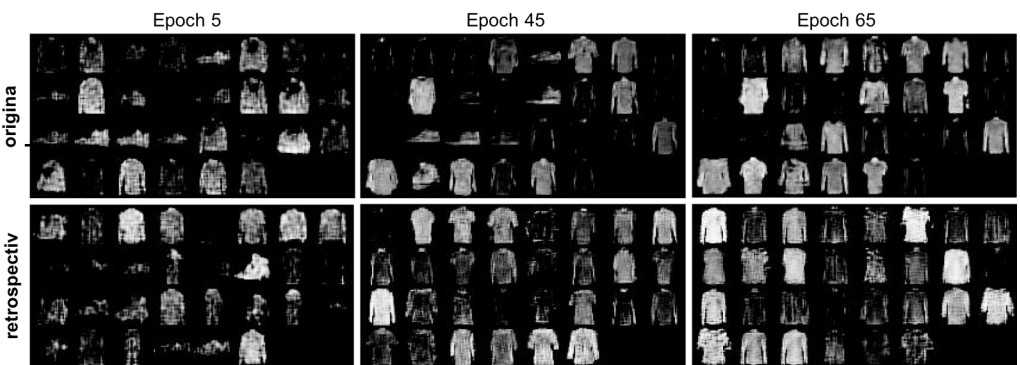

Figure 4: Images generated over training epochs when ACGAN (Odena et al., 2016) trained on FMNIST dataset: (a) without retrospection (row 1) (b) with retrospection (row 2)

For all experiments, the retrospection loss is initialized without any warm-up period (zero epochs). The loss is computed on outputs of the discriminator and is used to train the generator model. For DCGAN (Radford et al., 2015) and LSGAN (Mao et al., 2016) $L_2$-norm as choice of norm. The retrospective update happens six times in one epoch. The scaling parameter, $\kappa$ is initialized at 4 and is not changed during training. For ACGAN (Odena et al., 2016), which is conditional, the retrospective loss consists of both adversarial loss and class loss components. $L_1$-norm is used for class component and $L_2$-norm is used for adversarial component. Figure 3 presents comparative inception score plots when the various dataset-network pairs are trained with (without) the retrospection loss. Additionally, Figure 4 presents images generated over epochs when training ACGAN (Odena et al., 2016), with and without retrospection, on F-MNIST (Xiao et al., 2017).

### 4.3 SPEECH RECOGNITION

We perform speech recognition experiments using the Google Commands (Warden, 2017) dataset. The dataset consists of 65,000 utterances, where each utterance is about one-second long and belongs to one out of 30 classes. The classes correspond to voice commands such as yes, no, down, left, as pronounced by a few thousand different speakers. We follow (Zhang et al., 2017) to preprocess the utterances where we first extract normalized spectrograms from the original waveforms at a sampling rate of 16 kHz and subsequently we zero-pad the spectrograms to equalize their sizes at 160 x 101.

For this experiment, we compare LeNet(Lecun et al., 2001) and VGG-11(Simonyan & Zisserman, 2014) architecture, each of which is composed of two convolutional and two fully-connected layers. We train each model for 30 epochs with minibatches of 100 examples, using Adam as the optimizer. Training starts with a learning rate of $3\mathrm{x}10^-3$ and is divided by 10 every 10 epochs. The retrospective loss is introduced after a warm-up period of eight epochs, since we find it speeds up initial convergence. The retrospection update frequency is half epoch. The loss scaling margin, $\kappa$,

| Model | Validation Set | | Testing Set | |
|---|---|---|---|---|
| LeNet | original | 9.8 | original | 10.3 |
| | retrospective | **9.6** | retrospective | **9.9** |
| VGG-11 | original | 5.2 | original | 5.0 |
| | retrospective | **4.4** | retrospective | **4.2** |

Table 2: Classification error using retrospection on the Google Commands dataset

is initialized at 4, and is increased by 1% at each retrospective update. Results in Table 2 highlight that training using the retrospection loss decreases error rate for both LeNet (Lecun et al., 2001) and VGG-11 (Simonyan & Zisserman, 2014) on both validation and testing sets.

### 4.4 TEXT CLASSIFICATION

We perform text classification experiments on the task of emotion detection in dyadic conversations. We baseline our experiments against DialogueRNN (Majumder et al., 2019), a recent state-of-the-art work, which is composed of an attentive network consisting of three Gated Recurrent Units(GRU). We perform experiments using AVEC (Schuller et al., 2012) and IEMOCAP (Busso et al., 2008) datasets. While the datasets are multi-modal (image and text), following (Majumder et al., 2019), we restrict scope of our experiments to using text. To feed into the network, the text data is preprocessed to obtain n-gram features as detailed in (Majumder et al., 2019). We follow the same train-test split and training configurations as in the original work. Performance comparison is reported against $BiDialogueRNN_+Att$, the best performing variant from the original work.

For experiments on IEMOCAP, models in each experiment are trained for 60 epochs on cross-entropy objective with F1-Score and accuracy as performance metrics. For retrospection, a warm-up of zero epochs is used. On AVEC, models in each experiment are trained for 100 epochs using MSE loss with MSE and pear-score(r) as the performance metrics. Here, introducing the retrospection loss after a warm-up of sev-

| Dataset | IEMOCAP | | AVEC | |
|---|---|---|---|---|
| | Accuracy | F1-Score | MSE | Pear-Score (r) |
| original | 62.66 | 62.75 | 0.179 | 0.318 |
| retrospective | **64.60** | **64.75** | **0.177** | **0.332** |

Table 3: Performance on using retrospection on task of dyadic emotion recognition on DialogueRNN

enty five epochs produces best performance. For experiments on both IEMOCAP and AVEC, the retrospective update frequency is one epoch. The loss scaling margin, $\kappa$, is set to 4 at initialization and is updated by 2% at each retrospective update. Experiments are conducted using the official

code repository (Co, 2019). Results in Table 3 show that using the retrospection loss when training DialogueRNN improves performance on both IECOMAP and AVEC datasets.

## 4.5 Few-shot Classification

We conduct experiments on the task of few shot classification using the CUB-200 (Wah et al., 2011) dataset. The CUB-200 dataset consists of 11,788 images from 200 bird species. In few-shot learning, the ability of a model is measured by its performance on n-shot, k-way tasks where the model is given a query sample belonging to a new, previously unseen class and a support set, S, consisting of n examples each from k different unseen classes. The model then has to determine which of the support set classes the query sample belongs to. We restrict the scope of our experiments to the 5-way 5-shot setting and baseline against closerlook (Chen et al., 2019), a recent state-of-the-art work, and protonet (Snell et al., 2017) another popular work from the domain. Our experiments follow from (Chen et al., 2019) and implementations use code in (Chen, 2019). We conduct experiments with backbones of varying depths - Conv4, Conv6 and ResNet34, as presented in (Chen et al., 2019).

For our experiments, each model is trained on protonet (Snell et al., 2017) for 400 epochs and on closerlook (Chen et al., 2019) for 200 epochs. For Conv4 and Conv6 configurations on both closerlook and protonet, retrospection is introduced without any warm-up period (zero epochs).

| Model | protonet | | closerlook | |
|---|---|---|---|---|
| | original | retrospective | original | retrospective |
| Conv4 | 75.26 ±1.05 | **77.42 ± 1.25** | 79.03 ±0.63 | **79.95 ± 0.75** |
| Conv6 | 80.71 ±1.55 | **81.78 ± 1.40** | 81.05 ±0.55 | **81.35 ± 0.30** |
| ResNet34 | 88.75 ±1.01 | **89.99 ± 1.13** | 82.23 ±0.59 | **83.11 ± 0.55** |

Table 4: Classification performance using retrospection for few-shot classification on CUB dataset

For ResNet34, a warm-up period of 280 epochs for protonet and 150 epochs for closerlook is used. For all experiments, the retrospective update frequency is one epoch each. The scaling parameter, $\kappa$, is initialized at 4 and increased by 2% at each retrospective update. For closerlook, we report comparative performance with baseline++, the best performing variant. Results in Table 4 highlight that training with the retrospective loss results in improved classification accuracy for all backbones configurations on both closerlook and protonet. [1]

## 5 Analysis

In this section, we presents ablation studies to analyse the impact of different hyperparameters - batch size, optimizer, retrospective update frequency ($F$) and the scaling parameter $\kappa$. The studies are conducted on the task of image classification on F-MNIST (Xiao et al., 2017) dataset using LeNet (Lecun et al., 2001) architecture. The default training configurations are used from Section 4.1. In all the studies, networks trained for each configuration are initialized with the same weights to ensure consistent comparison.

**Impact of Batch Size** We perform experiments to analyse the invariance of the proposed retrospection loss to batch size. For this study, we consider batch sizes - 32, 64, 128. Results presented in Figure 5 highlight that the retrospection loss results in improved training performance, which is achieved much faster, across all the different batch sizes.

**Impact of Optimizer** We perform experiments to analyse the invariance of the proposed retrospection loss to choice of optimizer. For this study, we use Adam (Kingma & Ba, 2014) and SGD optimizers. The classification performance when using Adam and SGD (momentum=0.5) are reported in Figure 6 (Row 2). The observed results highlight that the retrospective loss results in improved training performance across different optimizers.

**Choice of Retrospective Update Frequency, $F$.** We study the impact of different update frequencies ($F$) for the retrospective loss. We experiment with 150, 200, 250 steps. Results are presented in Figure 6 (Row 1) with the best performance achieved using $F = 250$ steps. All configurations of the retrospection loss outperforms the configuration (in blue) trained without it. While experiments in the current work used randomized search to estimate update frequencies, retrospective mining can be an interesting future direction.

---

[1]Results in some experiments on the original configuration do not match values (are higher or lower) reported in (Chen et al., 2019) even after using official code and same training config. However, we ensure consistency of comparison by using the same initializations for original and retrospective configurations

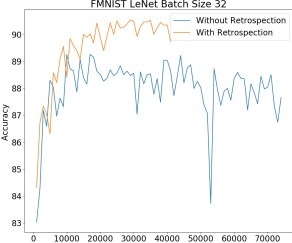 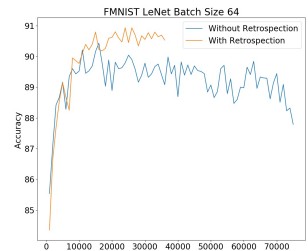 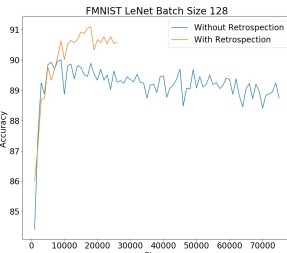

Figure 5: Classification performance using retrospection on LeNet(Lecun et al., 2001) across different batch sizes on FMNIST (Xiao et al., 2017)

**Choice of scaling margin,** $\kappa$ We conduct experiments using different initial values of the loss scaling margin, $\kappa$. For this analysis, the value of $\kappa$ remains unchanged during the training. Results are presented in Figure 6 (Row 1) with best performance achieved with $\kappa = 4$. All configurations produce better performance than with $\kappa = 1$.

# 6 CONCLUSION AND FUTURE WORK

In this work, we introduced a retrospective loss that utilizes parameter states from previous training steps to condition weight updates and guide the network towards convergence. We conduct experiments across multiple tasks, input types and architectures to empirically validate the effectiveness of the proposed loss. We perform ablation studies to analyze its behaviour. As an interesting future direction to explore the connection between retrospection and momentum, we conducted preliminary experiments on image classi-

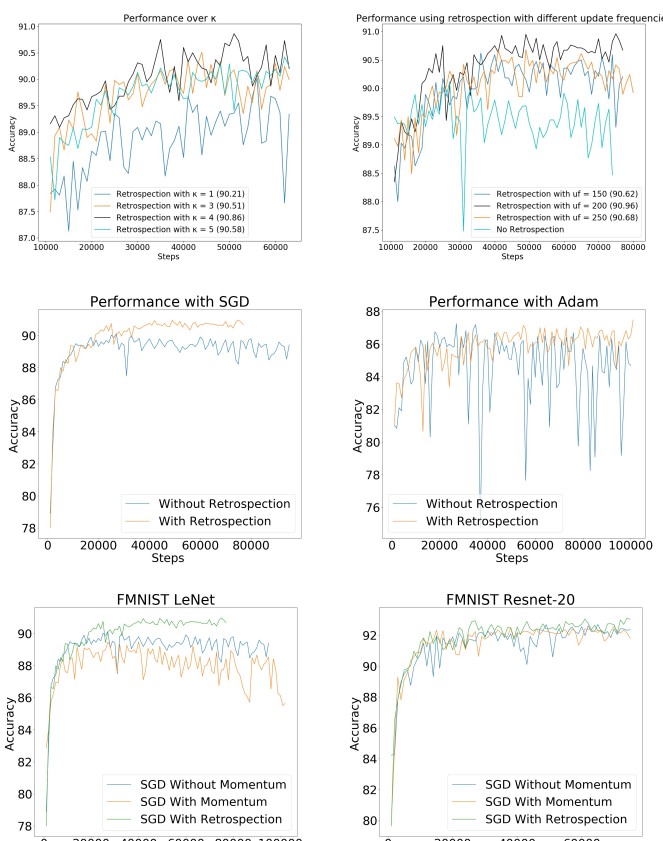

Figure 6: Ablation studies of classification performance on the F-MNIST dataset: *(Row 1)* Varying loss scaling parameter, $\kappa$ *(left)*, and retrospective update frequency *(right)*; *(Row 2)* Using retrospection on LeNet (Lecun et al., 2001) and SGD vs Adam optimizers; *(Row 3)* Using SGD, SGD +momentum, SGD + retrospection for LeNet, ResNet-20 architectures

fication to evaluate the impact of the retrospective loss on optimization. We contrast performance from three different configurations on image classification: (a) trained without retrospective loss (SGD); (b) trained without retrospective loss (SGD + momentum); and (c) with retrospective loss (SGD). Results in Figure 6 (Row 3) highlight that introducing retrospection improves performance (blue vs green); moreover, using the retrospective loss improves convergence even when SGD is optimized without momentum.

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

## A    EXPERIMENTAL RESULTS ON GRAPH NEURAL NETWORKS

We study the impact of the retrospection loss on the task of semi-supervised node classification using *CORA* and *CITESEER* datasets (Sen et al., 2008). For our experiments, we use two different models: ARMA (Bianchi et al., 2019) (a recent state-of-the-art method) and GCN (Kipf & Welling, 2016), another popular variant. Our implementations follow from (Fey & Lenssen, 2019). Performance is reported by averaging results over 30 experimental runs, each of which involves training the model for 100 epochs. For all experiments, the retrospective loss is introduced without any warm-up period (zero epochs). The hyperparameters, $F$ and $\kappa$, used for training on both datasets (CORA and CITESEER) for each of the three networks are: a) GCN: $F = 2$, $\kappa = 4$; b) ARMA: $F = 1$, $\kappa=3$. Table 5 presents quantitative impact of using the retrospection loss.

| Dataset | Config | GCN | ARMA |
|---------|--------|-----|------|
| CORA | original | 80.85 ±0.53 | 78.53 ±1.5 |
| | retrospective | **81.23 ± 0.27** | **79.45 ± 1.15** |
| CITESEER | original | 70.65 ±0.93 | 63.63 ±1.3 |
| | retrospective | **71.25 ± 0.75** | **64.22 ± 1.2** |

Table 5: Classification performance using retrospection loss on graph neural networks

## B    ROBUSTNESS OF RESULTS

To ensure consistency of comparison, we reported performance in the main paper by initializing both retrospective and original experiments with the same weights. Now, for comprehensive analysis, we report experimental values for Image Classification, Speech Recognition and Text Classification tasks averaged over 10 runs. (Note that for few-shot learning, we already included this information in the main paper.) Table 6, Table 7 and Table 8 present the corresponding mean and standard deviation of the results for image, text and speech classification respectively. We also note that all the results in the submitted paper are in the same range as the mean ± std in the results below, although these were separately performed - showing the consistency.

| Dataset | Network | Original | Retrospective |
|---------|---------|----------|---------------|
| SVHN | VGG-11 | 5.51 ±0.08 | **4.75 ± 0.09** |
| | ResNet-18 | 4.38 ±0.09 | **4.01 ± 0.07** |
| F-MINST | ResNet-20 | 7.63 ±0.04 | **6.85 ± 0.06** |
| | LeNet | 10.74 ±0.19 | **9.37 ± 0.14** |

Table 6: Mean and Standard Deviation of error rates for Image Classification

| Method | IECOMAP | | AVEC | |
|--------|---------|--|------|--|
| | F1-Score | Accuracy | MSE | R (Pear Score) |
| Retrospective | **64.40 ± 0.4** | **64.97 ± 0.5** | **0.1772 ± 0.0006** | **0.332 ± 0.008** |
| Original | 62.60 ±0.9 | 62.70 ±0.7 | 0.1798 ±0.0005 | 0.317 ±0.007 |

Table 7: Mean and Standard Deviation of results for Text Classification

| Model | Validation Set | | Testing Set | |
|-------|----------------|--|-------------|--|
| | Original | Retrospective | Original | Retrospective |
| LeNet | 9.77 ±0.05 | **9.60 ± 0.03** | 10.26 ±0.05 | **9.86 ± 0.04** |
| VGG-11 | 5.15 ±0.08 | **4.37 ± 0.04** | 5.03 ±0.06 | **4.16 ± 0.05** |

Table 8: Mean and Standard Deviation of error rates for Speech Recognition

## C    ABLATION STUDY: MOMENTUM

We also conducted an additional study to analyse the impact of choice of momentum values in the optimizer when using the retrospection loss in training. As in other ablation studies, we train LeNet for image classification on F-MNIST using SGD and experiment with different values of

the momentum parameter (0.5, 0.7, 0.9). The other parameter configurations remain the same as initially presented in Section 4.1 (lr=0.1, batch_size=32). As highlighted by results in Table 9, the retrospection loss is independent of momentum value since retrospective training results in better performance than original training (w/o retrospection) for all the different momentum values.

| Momentum = 0.5 | | Momentum = 0.9 | | Momentum = 0.7 | |
|---|---|---|---|---|---|
| Original | Retrospective | Original | Retrospective | Original | Retrospective |
| 10.8 | **9.4** | 10.05 | **9.06** | 9.51 | **8.94** |

Table 9: Impact of choice of momentum value on retrospection loss

## D    ABLATION STUDY: WARM-UP PERIOD

As in other ablation studies, we train LeNet on the task of FMNIST (60,000 images) image classification for 70,000 iterations with batch_size = 32 using SGD (mom=0.9). The error rates with different warm-ups are presented in Table 10. We observed that on simpler datasets (like FMNIST) since networks start at a reasonable accuracy, retrospection is effective even when we introduce it with a very low warm-up period ($I_w = 0$).

| Network | Original | Retrospective | | | |
|---|---|---|---|---|---|
| | $I_w = Infinity$ | $I_w = 0$ | $I_w = 10k$ | $I_w = 15k$ | $I_w = 20k$ |
| LeNet | 10.05 | 9.06 | 9.3 | 9.33 | 9.06 |

Table 10: For simple datasets like FMNIST, the retrospection loss is effective even when we introduce it with a low warm-up period. All different warm-up configurations improve performance over baseline.

Further, we observed that for tasks on more complex datasets with bigger networks, it is best to introduce the retrospection loss after training the network for some epochs when the network has started to converge to some extent, empirically around 50-75% of the training epochs. While introducing retrospection early also improves over baseline, later introduction of the retrospection loss further improves performance. Table 11 presents results obtained when we trained ResNet-56 on the task of image classification using CIFAR-10 for 200 epochs. Here, when the network is trained without retrospection (the original config as in the ResNet paper), we got an error rate of 6.86 (6.97 is reported in ResNet paper). However, on using retrospection, performance improved to 6.78 when the warm-up period ($I_w$) of 50 epochs was used and it further improved to 6.52 with a warm-up period of 150 epochs.

| Network | Original | Retrospective | | | |
|---|---|---|---|---|---|
| | $I_w = Infinity$ | $I_w$=0 | $I_w$=50 | $I_w$=100 | $I_w$=150 |
| ResNet-56 | 6.86 (6.97) | 6.81 | 6.78 | 6.61 | 6.52 |

Table 11: All the different warm-up configurations improve over baseline trained without retrospection loss. A higher warm-up period results in better performance when more complex datasets are trained using large networks.

Next, we study the impact of the warm-up period on the task of image generation using GAN's. For the GAN experiments in the main paper, we used the retrospection loss with a warm-up period of zero epochs which resulted in improved performance over the baselines. We believe that since GANs are inherently unstable and do not train to a fixed target, the warm-up period is unlikely to have a significant impact. Here, we present results from our study of the impact of different warm-up periods when training GANs. Figure 7 plots the inception scores when DCGAN is trained on FMNIST dataset using retrospection loss with different warm-up periods (0, 10 and 30 epochs) and all other parameters are same as in Section 4.2. As highlighted by the results, retrospection trained with all the three warm-up configurations improves upon the baseline method (better max inception scores) trained without retrospection (blue). For training with the retrospection loss, the warm-up period does not have a significant impact on overall performance with warm-up of 10 epochs (green) and 30 epochs (red) producing almost similar peak values.

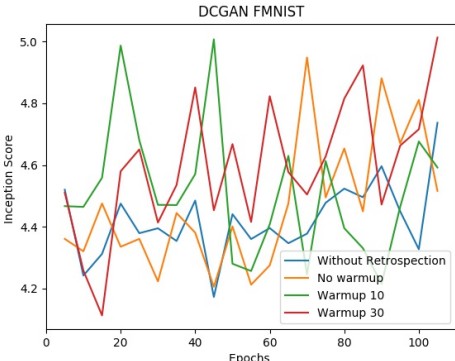

Figure 7: Training DCGAN on FMNIST dataset using retrospection loss with different warm-up periods.

## E   ABLATION STUDY: NORM

In this work, we seek to present the retrospection loss as a general concept that encourages using past parameter states as guidance during training to improve performance. Hence, we conduct an ablation study to analyse the effect of using different norms for the retrospection loss. Table 12 and Table 13 presents results of using retrospection loss with L1-norm and L2-norm on the task of image classification. As highlighted by the results, both configurations of the retrospection loss (L1-norm, L2-norm) improved performance as compared to training without retrospection but using L1-norm resulted in the better performance.

| Network | Original | L1-norm | L2-norm |
|---|---|---|---|
| LeNet | 10.8 | **9.4** | 9.7 |
| ResNet-20 | 7.6 | **6.8** | 7.3 |

Table 12: Error rates on using retrospection loss with different norms for Image Classification on F-MNIST dataset.

| Network | Original | L1-norm | L2-norm |
|---|---|---|---|
| ResNet-18 | 4.42 | **4.06** | 4.27 |
| VGG-11 | 5.54 | **4.70** | 5.15 |

Table 13: Error rates on using retrospection loss with different norms for Image Classification on SVHN dataset.

Further, we even tried a KL-divergence based formulation of the retrospection loss. Consider an input $(x_i, y_i)$ and network $g_\theta$ parameterized by $\theta$. Here $g_\theta(x_i)$ are the activations of the softmax layer and $y_i$ is the ground-truth class embedding. For the loss, we define: $out_{curr} = g_{\theta^T}(x_i)$ ; $out_{prev} = g_{\theta^{T_p}}(x_i)$ ; $target = y_i$. For KL-divergence, we used the following formulation of the retrospective loss at a training step T:

$$Loss(KL) = -1 * KL\_div(out_{curr}, out_{prev}) + CrossEntropy(out_{curr}, target) \qquad (4)$$

In the above experiment on SVHN, we obtained 5.45 and 4.31 as error rates for VGG-11 and ResNet-18 respectively.

While all our variants, L1-norm, L2-norm and KL-divergence, improved upon baselines, L1-norm resulted in better performance across tasks, except in unconditional GANs, where L2-norm is used to apply the retrospective loss on the adversarial predictions of the generator (Sec 4.2). One hypothesis is that when the L1-norm is used, the gradient is simply a dimension-wise sign (+ve vs -ve), which provides a clearer direction to gradient updates, especially when training to a fixed target embedding in predictive tasks.

