# OpenReview forum: "Retrospection: Leveraging the Past for Efficient Training of Deep Neural Networks"
_ICLR.cc/2020/Conference — Reject_

### Official Review · AnonReviewer1 · 2019-10-22
**Official Blind Review #1**

**Rating:** 8

**Review:**

The paper proposes a new loss function which adds to the training objective another term that pulls the current parameters of a neural network further away from the parameters at a previous time step.
Intuitively, this aims to push the current parameters further to the local optimum.
On a variety of benchmarks, optimizing the proposed loss function achieves better results than just optimizing the training loss.

The paper is well written and easy to follow.  However, I am not entirely convinced about the intuition of the proposed method and I think further investigation are necessary.
While the method is simple and general, it also seems to be rather heuristic and requires carefully chosen hyperparameters.
Having said that, the empirical evidence shows that the proposed loss function consistently improves performance.
The following details should be addressed further:

- I am a bit confused by the definition of the loss function. In Equation 1 it seems that the term on the left represents the training objective. If that is correct than Equation 2 second case contains the training objective twice?

- F in Section 3 after Equation 2 is not properly defined

- Could it happen that the proposed loss function leads to divergence, for example if the parameter from a previous time step theta^Tp is close to the optimum theta_star?

- What is the motivation to use the L1 norm? How does this choice affect convergence compared to let's L2 norm?

- Section 4.1 typo in first paragraph: K instead of \kappa

- Section 4.1 the results would be more convincing if all networks were trained multiple times with a different random initialization and Table 1 would include the mean and std.

- Why is no warm-up period used for the GAN experiments?

- Section 4.3: why is \kappa increase by 1% for the speech recognition experiments where as by 2% for all other experiments?

- I suggest to increase the line width of all figures since they are somewhat hard to identify on a print version.

- Why is the momentum set to 0.5 for SGD in the ablation study? Most frameworks use a default value of 0.9.

- I would like to see the affect of the warm-up period to the performance in the ablation study.

- How does the choice of learning rate schedule, such as for example cosine annealing, affect the loss function?



post rebuttal
------------------

I thank the authors for clarifying my questions and providing additional experiments. I think that especially the additional ablation studies and reporting the mean and std of multiple trials make the contribution of the paper more convincing. Hence, I increased my score.

**Experience Assessment:**

I have read many papers in this area.

**Review Assessment: Checking Correctness Of Derivations And Theory:**

N/A

**Review Assessment: Checking Correctness Of Experiments:**

I carefully checked the experiments.

**Review Assessment: Thoroughness In Paper Reading:**

N/A

---

> ### Author Response · Authors · 2019-11-09
> **Response for Reviewer #1 (Part 1/4)**
>
> We thank the reviewer for the detailed review and helpful comments. The comments were insightful in recommending further experiments, which we have described below and also added to the appendix of the revised draft. We have also updated the paper by incorporating the recommended corrections.
>
> Before we address the specific questions, we summarize our key contributions below for clarity:
> 1. We propose a new “retrospective loss” that is based on looking back at the trajectory of gradient descent and providing an earlier parameter state as guidance for further learning.
> 2. The key benefits of the proposed loss are that - it is simple and easy to implement (with any existing loss). Its simplicity allows us to easily generalize its use across tasks and application domains.
> 3. Our exhaustive experiments on a wide range of tasks including image classification (+ few-shot learning), GANs, speech recognition, text classification, and graph classification (included in the reply here) beat state-of-the-art methods on benchmark datasets with the addition of this loss term.
> 4. To the best of our knowledge, this is the first such effort; our empirical studies showed a consistent improvement in performance across the tasks in our multiple trials, demonstrating the potential of this method to have a strong impact on practical use in real-world applications across domains.
>
> We have also added these at the end of Section 1 in our updated paper.
>
> Below is our response to the individual comments (Q = question; R = our response):
>
> Q: “...concerned about the intuition of the proposed method…”
> R: The summary of the contributions above answers this question partially. The intuition for the proposed method comes from the observation that the past parameter states (with the timestep chosen reasonably) are generally more erroneous than the current state, and contain valuable information on where the weight update should not go.
> This information/direction is fully available with us during training but left unused. Our method leverages this information to provide a way to improve training performance. This can be more difficult in SGD (due to the noisy trajectory of updates) than in Batch GD, but we found choosing the timestep to not be difficult even when implementing this for SGD. Note that our approach is different from momentum, which we discuss in Sec 3 and Fig 6. From a different perspective, one can also view the proposed method as replicating the success of triplet loss in the data space, now in the parameter space (as discussed in Sec 3 of the paper).
>
> =============================================
> Q: “...carefully tuning hyperparameters…”
> R: We note that we did not need much careful tuning of hyperparameters for the results presented in this paper. The presented ablation studies and subsequent discussions (answered later) highlight that the proposed loss function outperforms baselines across diverse hyperparameter configurations (momentum, LR schedule, batch_size, etc..) without modifying the retrospective hyperparameters. Hyperparameter tuning to further boost performance was, in fact, for future work (as mentioned in Sec 3)
> We arrived at this formulation after careful deliberation and empirical studies, and the method is well-motivated as shown by the intuitions above. Importantly, the method’s simplicity and ease of implementation allow us to consistently get better performance than state-of-the-art methods across multiple tasks, as mentioned in the summary above - with potential for practical applicability in real-world applications.
>
> =============================================
> Q: “...confused by loss function...Eqn 1 - term on left represents training objective...in Eqn 2, the second case contains the training objective twice?…”
> R: We introduce the retrospection loss as an additional objective to be used alongside the original task-specific loss to improve training performance. If the task loss is also an Lp-norm, then this might be the case. This was however not the case in any of our experiments or tasks. We note that we also have the \kappa coefficient in Eqn 1, which allows us to control the influence of this term if this becomes required in such a case, consistent with the geometric intuition in Fig 1.
>
> =============================================
> Q: “...F in Section 3 after Equation 2 is not properly defined…”
> R: Thanks for pointing this out. F is the retrospective update frequency, we have added this to the revised submission.
>
> =============================================
> ***RESPONSE CONTINUED IN PART 2/4***

---

> > ### Author Response · Authors · 2019-11-09
> > **Response for Reviewer #1 (Part 2/4)**
> >
> > **CONTINUED FROM PART 1/4****
> >
> > Q: “...Could it happen that the proposed loss function leads to divergence...?”
> > R: As the training proceeds, \theta^{T_p} is more likely to become closer to \theta^{*} in later stages. In this case, the proposed formulation of the retrospection loss, where \kappa increases with training steps, increases the penalty on the divergence of the current state from the optimal parameter state (left-term of the loss) which helps facilitate convergence (please see the line in Sec 3 before Fig 1).
> >
> > In general, the recursive formulation of the retrospective loss helps in improving convergence. For instance, in case \theta^{T_p}, a previous parameter state diverges, in subsequent training steps, the retrospective loss will push \theta^T (the current state at step T) away from \theta^{T_p} and towards \theta^{*} (the optimal state), helping restore the balance.
> >
> > We agree that there are cases where the proposed method can diverge. However, from our experiments and studies, we found that the proposed loss function almost always helps in better convergence in practice. If the update frequency is very large, this could potentially lead to lack of convergence. Considering this can however be controlled using a hyperparameter (which is only one of two hyperparams in our entire framework), this issue is easily overcome.
> > =============================================
> >
> > Q: “...motivation to use the L1 norm? ...choice affect convergence compared to L2 norm..?”
> > R: Our framework is independent of the norm, and we in fact experimented with other norms in our experiments. We have included the results with L1 and L2 norms below. L1 norm provided the best results overall and hence was presented in the paper. Considering the choice of L1 norm is an implementation detail, we have moved the sentence regarding L1-norm in methodology to the experiments section.
> >
> > IMAGE CLASSIFICATION TEST ERROR ON F-MNIST (Lower the better)
> > —————————————————————
> > Network     || Original || L1-norm || L2-norm
> > —————————————————————
> > LeNet         ||    10.8   ||     9.4     ||     9.7
> > ResNet-20  ||    7.6     ||     6.8     ||     7.3
> > —————————————————————
> >
> > IMAGE CLASSIFICATION TEST ERROR ON SVHN (Lower the better)
> > —————————————————————
> > Network    || Original || L1-norm || L2-norm
> > —————————————————————
> > VGG-11     ||    5.54   ||     4.70   ||     5.15
> > ResNet-18 ||    4.42   ||     4.06   ||     4.27
> > —————————————————————
> >
> > We, in fact, even tried a KL-divergence based formulation of the retrospection loss. Consider an input (x_i, y_i) and network G_{\theta} parameterized by \theta. Here G_{\theta}(x_i) are the activations of the softmax layer and y_i is the ground-truth class embedding. For the loss, we define: output_curr = G_{\theta^T}(x_i) ; output_prev = G_{\theta^T_p}(x_i) ; target = y_i. For KL_div, we used the following formulation of the retrospective loss at a training step T: Loss(KL) = -1*KLDiv(output_curr, output_prev) + CrossEntropy(output_curr, target). In the above experiment on SVHN, we obtained 5.45 and 4.31 as error rates for VGG-11 and ResNet-18 respectively.
> > We have added these results to appendix E.
> >
> > While all our variants, L1-norm, L2-norm and KL_div, improved upon baselines, L1-norm resulted in better performance across tasks, except in unconditional GANs, where L2-norm is used to apply the retrospective loss on the adversarial predictions of the generator (Sec 4.2). One hypothesis for the outperformance of L1-norm is that when the L1-norm is used, the gradient is simply a dimension-wise sign (+ve vs -ve), which provides a clearer direction to gradient updates, especially when training to a fixed target embedding in predictive tasks.
> > =============================================
> >
> > Q: “...results … include the mean and std...”
> > R: We did run multiple trials during our studies, and our results in the paper were consistent across these trials. We, however, ran the experiments again, and are reporting our results below for Image Classification, Speech Recognition and Text Classification tasks averaged over 10 runs. (Note that for few-shot learning, we already included this information in the original submission). We also note that all the results in the submitted paper are in the same range as the mean +- std in the results below, although these were separately performed - showing the consistency.
> >
> > IMAGE CLASSIFICATION TEST ERROR (Lower the better)
> > ——————————————————————————-
> > Dataset     ||  Network       ||     original      ||  Retrospective
> > ——————————————————————————-
> > SVHN       ||   VGG-11      || 5.51 +- 0.08  ||  4.75 +- 0.09
> > SVHN       ||   ResNet-18  || 4.38 +- 0.09  ||  4.01 +- 0.07
> > F-MNIST  ||   LeNet          ||10.74 +- 0.19 ||  9.37 +- 0.14
> > F-MNIST  ||   ResNet-20  || 7.63 +- 0.04  ||  6.85 +- 0.06
> > ——————————————————————————-
> >
> > ***RESPONSE CONTINUED IN PART 3/4***

---

> > > ### Author Response · Authors · 2019-11-09
> > > **Response for Reviewer #1 (Part 3/4)**
> > >
> > > **CONTINUED FROM PART 2/4****
> > >
> > > TEXT CLASSIFICATION TEST ERROR (H = Higher the better, L = lower the better)
> > > ——————————————————————————————————————---
> > >     Method       ||               IECOMAP                ||                        AVEC
> > > ———————————————————————————————————————--
> > >                        || F1-Score (H)   || Accuracy  (H)  ||        MSE     (L)    ||   r (Pear Score) (H)
> > > ———————————————————————————————————————-
> > > Retrospective || 64.40 +- 0.4    ||  64.97 +- 0.5    ||  0.1772 +- 0.0006  ||  0.332 +- 0.008
> > > Original           || 62.60 +- 0.9    ||  62.70 +- 0.7    ||  0.1798 +- 0.0005  ||   0.317 +- 0.007
> > > ——————————————————————————————————————--
> > >
> > > SPEECH CLASSIFICATION ERROR RATES (Lower the better)
> > > —————————————————————————————————————-
> > > Network  ||                   Validation Set             ||                   Testing Set
> > > —————————————————————————————————————-
> > >                         Original        Retrospective     ||      Original                Retrospective
> > > —————————————————————————————————————-
> > > LeNet     ||    9.77 +- 0.05     9.60 +- 0.03       ||     10.26 +- 0.05        9.86 +- 0.04
> > > VGG-11  ||    5.15 +- 0.08     4.37 +- 0.04       ||     5.03 +- 0.06          4.16 +- 0.05
> > > —————————————————————————————————————-
> > > As an extension, we also did experiments on the task of node classification using Graph Neural Networks, where the performance is ideally reported by averaging over several runs. Here, we report performances by averaging over 30 runs each, where each run was trained for 100 epochs. We carried out experiments on CORA and CITESEER datasets using two widely used/state-of-the-art networks: ARMA (Bianchi et al., CoRR 2019), and GCN (Kipf & Welling, ICLR 2017). Using retrospection loss improves both accuracy and std. deviation in almost all cases.
> > > These experiments with details have been added to Appendix A of the revised paper.
> > >
> > > GRAPH NODE CLASSIFICATION TEST ACCURACY (Higher the better)
> > > ———————————————————————————————————
> > > Dataset       ||               GCN                          ||               ARMA
> > > ———————————————————————————————————
> > >                    ||      Original        Retrospective  ||      Original       Retrospective
> > > ———————————————————————————————————
> > > CORA         ||  80.85 +- 0.53    81.23 +- 0.27  ||  78.53 +- 1.5    79.45 +- 1.15
> > > CITESEER  ||  70.65 +- 0.93    71.25 +- 0.75  || 63.63 +- 1.3    64.22 +- 1.2
> > > ———————————————————————————————————
> > >
> > > =============================================
> > > Q: “... the effect of warm-up period ...in ablation study…”
> > > R:. As in the ablation studies, we trained LeNet on the F-MNIST dataset (60k images) for 70k iterations with batch_size = 32 (we use momentum=0.9). Hence, 1 epoch lasts for around 2k iterations. The error rates with different warm-ups are mentioned in the table below. We observed that on simpler datasets (like FMNIST), since networks start training at a reasonable accuracy, retrospection is effective even when we introduce it with a very low warm-up period (Iw = 0 steps).
> > >
> > > ABLATION STUDY: DIFFERENT WARM-UP PERIOD (Lower the better)
> > > ————————————————————————————
> > >         Original    ||                     Retrospective
> > > ————————————————————————————
> > >   Iw = Infinity   ||    Iw= 0  ||  Iw=10k || Iw = 15k || Iw = 20k
> > > ————————————————————————————
> > >       10.05           ||    9.06   ||      9.3      ||    9.33     ||   9.06
> > > ————————————————————————————
> > >
> > > Further, we observed that for tasks on more complex datasets, it is best to introduce the retrospection loss after training the network for some epochs when the network has started to converge to some extent, empirically around 50-75% of the training epochs. While introducing retrospection early also improves over the baseline, later introduction of the retrospection loss further improves performance. For instance, we trained ResNet-56 on the task of image classification using CIFAR-10 dataset for 200 epochs. Here, when the network is trained without retrospection (the original config as in the ResNet paper), we got an error rate of 6.86 (6.97 is reported in ResNet paper). However, on using retrospection, performance improved to 6.78 when the warm-up period (Iw) of 50 epochs was used and it further improved to 6.52 with a warm-up period of 150 epochs.
> > >
> > > IMAGE CLASSIFICATION ERROR RATE FOR RESNET-56 (CIFAR-10) (Lower the Better)
> > > ——————————————————————————
> > >         Original         ||                     Retrospective
> > > ——————————————————————————
> > >      Iw = Infinity     ||    Iw = 0    Iw= 50   Iw = 100    Iw= 150
> > > ——————————————————————————
> > >      6.86 (6.97)       ||      6.81         6.78        6.61         6.52
> > > ——————————————————————————
> > >
> > > =============================================
> > >
> > > ***RESPONSE CONTINUED IN PART 4/4***

---

> > > > ### Author Response · Authors · 2019-11-09
> > > > **Response for Reviewer #1 (Part 4/4)**
> > > >
> > > >
> > > > Q:” … \kappa increase by 1% for the speech recognition ... by 2% for all other...?”
> > > > R: In the paper, we reported the best performance for all experiments by doing a randomized search for hyperparameter values. While increasing kappa by 2\% at each retrospective update improved performance, we reported values for 1\% since it provided marginally better performance. In the table below, we compare error rates when \kappa is increased by 1\% vs 2\%. For consistent comparison, parameters in all experiments were restored from the same initialization. To give some context of improvement, we have also reported error rates when training without retrospection (Original).
> > > >
> > > > SPEECH RECOGNITION ERROR RATE WITH DIFFERENT \kappa CONFIGS (Lower the Better)
> > > > —————————————————————————————--
> > > > Network  ||           Validation Set        ||              Testing Set
> > > > —————————————————————————————--
> > > >                ||    Original     1\%     2\%   ||     Original     1\%         2\%
> > > > —————————————————————————————--
> > > > LeNet     ||     9.8          9.6       9.6     ||     10.3           9.9         10.0
> > > > VGG-11  ||     5.2          4.4       4.5     ||     5.0             4.2          4.4
> > > > ——————————————————————————————
> > > >
> > > > ============================
> > > >
> > > > Q: “...momentum set to 0.5 for SGD in the ablation study?... most...default value of 0.9…”
> > > > R: Thank you for pointing out this issue. Since the ablation studies were intending to show robustness to other hyperparameters (batch_size, F, k), we did not explicitly tune for the best momentum values and went with 0.5.
> > > > Now, we are presenting the same ablation study with different momentum values. We observe our approach is independent of the choice of momentum value since that retrospective training is better than the original training for all the different momentum values. The results of our experiments on FMNIST (as in Fig 6 of paper) are shown below.
> > > >
> > > > ABLATION STUDY: DIFFERENT MOMENTUM FOR SGD (Lower the Better)
> > > > ———————————————————————————————————-----
> > > >        Momentum=0.5          ||        Momentum = 0.9         ||     Momentum = 0.7
> > > > ————————————————————————————————————--
> > > >  Original    Retrospective  ||    Original     Retrospective  ||  Original     Retrospective
> > > > ———————————————————————————————————-—-
> > > >      10.8             9.4           ||    10.05              9.06            ||     9.51             8.94
> > > > ————————————————————————————————————---
> > > >
> > > > =================================
> > > >
> > > > Q:”...choice of learning rate schedule, ...example cosine annealing, affect the loss...?”
> > > > R:  Thanks for suggesting this experiment. We studied the impact of using cosine annealing LR schedule on the task of Image Classification and Speech Recognition when training with retrospection loss. In both cases, using retrospection loss still improves upon baselines that are trained without retrospection showing that the proposed loss is independent of the type of LR scheduler.
> > > >
> > > > IMAGE CLASSIFICATION ERROR RATE (FMNIST) WITH COSINE ANNEALING (Lower the Better)
> > > > —————————————————---
> > > > Network     ||  Original  ||  Retrospection
> > > > —————————————————---
> > > > LeNet         ||     9.71     ||        8.20
> > > > ResNet-20  ||    7.09      ||        6.44
> > > > —————————————————--
> > > >
> > > > SPEECH RECOGNITION ERROR RATE WITH COSINE ANNEALING (Lower the Better)
> > > > ———————————————————————————————
> > > > Network  ||            Validation Set           ||                   Testing Set
> > > > ———————————————————————————————
> > > >                      Original     Retrospective   ||    Original    Retrospective
> > > > ———————————————————————————————
> > > > VGG-11  ||      5.24            4.88               ||    5.73            5.01
> > > > LeNet     ||      9.94            9.63               ||    10.48          10.04
> > > > ———————————————————————————————
> > > >
> > > > Further, the various techniques we compare against on several diverse tasks ranging from image classification, speech recognition, text classification, few-shot classification, etc. follow distinct LR schedules, as mentioned in our experimental settings, and the proposed method results in performance improvement irrespective of the disparate configurations
> > > >
> > > > =========================
> > > >
> > > > Q:”...increasing the line width of all figures … somewhat hard to identify on a print version…”
> > > > R: Thanks a lot for suggesting this, we will incorporate this change in the camera-ready.
> > > > ======
> > > >
> > > > Q: “..no warm-up period used for the GAN experiments?..”
> > > > R: We believe that since GANs are inherently unstable and do not train to a fixed target, the warm-up period is unlikely to have an impact. Hence, we reported experiments in our original submission with a warm-up period of 0 epochs which resulted in performance improvement (max IS value) by faster convergence (Fig 3, Fig 4). Now, we have included an ablation study on the impact warm-up period on GAN training in appendix D, which corroborates the hypothesis.
> > > >
> > > > =====
> > > > Thank you again for your comments, and we will be happy to discuss further any clarifications/questions.

---

### Official Review · AnonReviewer2 · 2019-10-23
**Official Blind Review #2**

**Rating:** 3

**Review:**

This paper presents the retrospective loss to optimize neural network training. The idea behind the retrospective loss is to add a penalization term between the current model to the model from a few iterations before. Extensive experimental results on a wide range of datasets are provided to show the effectiveness of the retrospective loss.

The retrospective loss is additionally controlled by two hyperparameters, the strength parameter K and the update frequency T_p. This loss, measured in L-1 norm, is added to the training objective. The geometric intuition of the added loss term is that this pushes the model away from the model at iteration T_p. The paper argues that this shrinks the parameter space of the loss function.

One of the concern regards the writing of the paper.
- Algorithm 1 and Figure 6 look very blurry, which I think are both below the publication standard.
- The introduction could be written to be more helpful, such as providing more context on why the obtained experimental results are important (e.g. getting state-of-the-art results on the datasets studied in the experiments)
- The Related Work contrasts with previous work which is not clear because the precise contribution has not been stated at the point.

More detailed questions:
- What are the standard deviations for the experimental results (as you reported in Table 4 but not in other experiments)?
- I'm curious whether the use of L-1 norm is critical or not in the retrospective loss.

**Experience Assessment:**

I do not know much about this area.

**Review Assessment: Checking Correctness Of Derivations And Theory:**

I assessed the sensibility of the derivations and theory.

**Review Assessment: Checking Correctness Of Experiments:**

I assessed the sensibility of the experiments.

**Review Assessment: Thoroughness In Paper Reading:**

I made a quick assessment of this paper.

---

> ### Author Response · Authors · 2019-11-09
> **Response for Reviewer #2 (Part 1/2)**
>
> We thank the reviewer for the review and helpful comments. We value every feedback and have updated the draft to address these concerns too. We hope that the editorial suggestions (which have now been addressed) will not be held against the technical merit of the paper.
>
> Some specific changes to the paper are:
> a. Updates
>      1. The introduction is updated to highlight our contributions more explicitly.
>      2. Algorithm 1 and Figure 6 replaced with high-res variants.
> b.  Additions (in appendix)
>      1. Additional experiments on graph-structured data (with mean and std)
>      2. Mean and std deviation of current experiments
>      3. Ablation Study: Momentum for SGD
>      4. Ablation Study: Warm-up period
>      5. Ablation Study: Choice of Norm
>
> Before we address the specific questions, we summarize our key contributions below for clarity:
> 1. We propose a new “retrospective loss” that is based on looking back at the trajectory of gradient descent and providing an earlier parameter state as guidance for further learning.
> 2. The key benefits of the proposed loss are that - it is simple and easy to implement (with any existing loss). Its simplicity allows us to easily generalize its use across tasks and application domains.
> 3. Our exhaustive experiments on a wide range of tasks including image classification (+ few-shot learning), GANs, speech recognition, text classification, and graph classification (included here) beat state-of-the-art methods on benchmark datasets with the addition of this loss term.
> 4. To the best of our knowledge, this is the first such effort; our empirical studies showed a consistent improvement in performance across the tasks in our multiple trials, demonstrating the potential of this method to have a strong impact on practical use in real-world applications across domains.
>
> We have also added these at the end of Section 1 in our updated paper.
>
> Below is our response to the individual comments (Q = question; R = our response):
>
> Q: ...standard deviations for results…?
> R: We did run multiple trials during our studies, and our results in the paper were consistent across these trials. We, however, ran the experiments again, and are reporting our results below for Image Classification, Speech Recognition and Text Classification tasks averaged over 10 runs. (Note that for few-shot learning, we already included this information in the original submission). We also note that all the results in the submitted paper are in the same range as the mean +- std in the results below, although these were separately performed - showing the consistency.
>
> IMAGE CLASSIFICATION TEST ERROR (Lower the better)
> ——————————————————————————-
> Dataset     ||  Network       ||     original      ||  Retrospective
> ——————————————————————————-
> SVHN       ||   VGG-11       || 5.51 +- 0.08  ||  4.75 +- 0.09
> SVHN       ||   ResNet-18   || 4.38 +- 0.09  ||  4.01 +- 0.07
> F-MNIST  ||   LeNet          ||10.74 +- 0.19 ||  9.37 +- 0.14
> F-MNIST  ||   ResNet-20   || 7.63 +- 0.04  ||  6.85 +- 0.06
> ——————————————————————————-
>
> TEXT CLASSIFICATION TEST ERROR (H = Higher the better, L = lower the better)
> ——————————————————————————————————————---
>     Method       ||             IECOMAP                  ||                   AVEC
> ———————————————————————————————————————-
>                        || F1-Score (H)   || Accuracy  (H)  ||        MSE     (L)    ||   r (Pear Score) (H)
> ———————————————————————————————————————-
> Retrospective || 64.40 +- 0.4    ||  64.97 +- 0.5    ||  0.1772 +- 0.0006  ||  0.332 +- 0.008
> Original           || 62.60 +- 0.9    ||  62.70 +- 0.7    ||  0.1798 +- 0.0005  ||   0.317 +- 0.007
> ——————————————————————————————————————--
>
> SPEECH CLASSIFICATION ERROR RATES (Lower the better)
> —————————————————————————————————————-
> Network  ||                   Validation Set             ||                   Testing Set
> —————————————————————————————————————-
>                         Original        Retrospective     ||      Original                Retrospective
> —————————————————————————————————————-
> LeNet     ||    9.77 +- 0.05     9.60 +- 0.03       ||     10.26 +- 0.05        9.86 +- 0.04
> VGG-11  ||    5.15 +- 0.08     4.37 +- 0.04       ||     5.03 +- 0.06          4.16 +- 0.05
> —————————————————————————————————————-
>
> As an extension, we also did experiments on the task of node classification using Graph Neural Networks, where the performance is ideally reported by averaging over several runs. Here, we report performances by averaging over 30 runs each, where each run was trained for 100 epochs. We carried out experiments on CORA and CITESEER datasets using two widely used/state-of-the-art networks: ARMA (Bianchi et al., CoRR 2019), and GCN (Kipf & Welling, ICLR 2017). Using retrospection loss improves both accuracy and std. deviation in almost all cases.
> These experiments with details have been added to Appendix A of the revised paper.
>
> Due to space constraints, the response is continued in PART 2/2

---

> > ### Author Response · Authors · 2019-11-09
> > **Response for Reviewer #2 (Part 2/2)**
> >
> > ***CONTINUED FROM PART 1/2***
> >
> > GRAPH NODE CLASSIFICATION TEST ACCURACY (Higher the better)
> > ———————————————————————————————————
> > Dataset       ||               GCN                          ||               ARMA
> > ———————————————————————————————————
> >                    ||      Original        Retrospective  ||      Original       Retrospective
> > ———————————————————————————————————
> > CORA         ||  80.85 +- 0.53    81.23 +- 0.27  ||  78.53 +- 1.5    79.45 +- 1.15
> > CITESEER  ||  70.65 +- 0.93    71.25 +- 0.75  || 63.63 +- 1.3    64.22 +- 1.2
> > ———————————————————————————————————
> >
> > Q: I'm curious whether the use of the L1 norm is critical or not in the retrospective loss.
> > R: Our framework is independent of the norm, and we in fact experimented with other norms in our experiments. We have included the results with L1 and L2 norms below. L1 norm provided the best results overall, and hence was presented in the paper. Considering the choice of L1 norm is an implementation detail, we have moved the sentence regarding L1-norm in methodology to the experiments section.
> >
> > IMAGE CLASSIFICATION TEST ERROR ON F-MNIST (Lower the better)
> > ——————————————————————
> > Network     || Original || L1-norm || L2-norm
> > ——————————————————————
> > LeNet         ||    10.8   ||     9.4      ||     9.7
> > ResNet-20  ||    7.6     ||     6.8      ||     7.3
> > ——————————————————————
> >
> > IMAGE CLASSIFICATION TEST ERROR ON SVHN (Lower the better)
> > —————————————————————
> > Network    || Original || L1-norm || L2-norm
> > —————————————————————
> > VGG-11     ||    5.54   ||     4.70    ||    5.15
> > ResNet-18 ||    4.42   ||     4.06   ||    4.27
> > —————————————————————
> >
> > We, in fact, even tried a KL-divergence based formulation of the retrospection loss. Consider an input (x_i, y_i) and network $G_{\theta}$ parameterized by $\theta$. Here $G_{\theta}(x_i)$ are the activations of the softmax layer and y_i is the ground-truth class embedding. For the loss, we define: output_curr = $G_{\theta^T}(x_i)$ ; output_prev = $G_{\theta^T_p}(x_i)$ ; target = y_i. For KL_div, we used the following formulation of the retrospective loss at a training step T: Loss(KL) = -1*KLDiv(output_curr, output_prev) + CrossEntropy(output_curr, target). In the above experiment on SVHN, we obtained 5.45 and 4.31 as error rates for VGG-11 and ResNet-18 respectively.
> > We have added these results to the Appendix E.
> >
> > While all our variants, L1-norm, L2-norm and KL_div, improved upon baselines, L1-norm resulted in better performance across tasks, except in unconditional GANs, where L2-norm is used to apply the retrospective loss on the adversarial predictions of the generator (Sec 4.2). One hypothesis is that when the L1-norm is used, the gradient is simply a dimension-wise sign (+ve vs -ve), which provides a clearer direction to gradient updates, especially when training to a fixed target embedding in predictive tasks.
> >
> > Thank you again for your comments, and we will be happy to discuss further for any clarifications/questions.

---

### Decision · Program_Chairs · 2019-12-19

**Decision:**

Reject

**Comment:**

This paper introduces a further regularizer, retrospection loss, for training neural networks, which leverages past parameter states.  The authors added several ablation studies and extra experiments during the rebuttal, which are helpful to show that their method is useful. However, this is still one of those papers that essentially proposes an additional heuristic to train deep news, which is helpful but not clearly motivated from a theoretical point of view (despite the intuitions). Yes, it provides improvements across tasks but these are all relatively small, and the method is more involved. Therefore, I am recommending rejection.